# Health-Seeking Behaviors and Misconceptions about Osteoarthritis in Patients and the General Population in Saudi Arabia

**DOI:** 10.3390/healthcare11091208

**Published:** 2023-04-23

**Authors:** Ambreen Kazi, Hamad F. Alrabiah, Khalid Fawaz Alosaimi, Naif Ahmed Alshehri, Omar Mohammad Bassam Alhalabi, Abdulelah Saad Alshamrani, AlJohara M AlQuaiz, Bushra Hamid

**Affiliations:** 1Princess Nora Bent Abdullah Chair for Women’s Health Research, Research Chairs Program, King Saud University Medical City, Riyadh 11362, Saudi Arabia; 2Department of Family & Community Medicine, College of Medicine, King Saud University Medical City, Riyadh 11362, Saudi Arabia; 3College of Medicine, King Saud University Medical City, Riyadh 11362, Saudi Arabia

**Keywords:** osteoarthritis, misconceptions, health-seeking behaviors, Saudi Arabia

## Abstract

Osteoarthritis (OA) is a public health disease that causes decreased mobility and leads to poor quality of life. A person’s health-seeking behavior can influence their understanding of a disease, which in turn can alter its course. The objectives of this study were to measure the misconceptions about osteoarthritis and to identify the associated health-seeking behaviors. An online, self-administered, questionnaire-based study was conducted with 872 Arabic-speaking participants divided into three strata, group 1 comprising of patients with OA, group 2 participants with joint pain (without OA) and group 3 comprised of general population. Multivariate logistic regression analysis found that seeking care from general practitioners [3.29 (1.19, 9.16)], taking advice from friends [2.83 (1.08, 7.42)], seeking care from chiropractors [3.67 (1.02, 13.60)] and podiatrist [4.64 (1.31, 16.51)] were significantly associated with misconceptions, whereas, the odds were lower for those using social media [0.16 (0.06, 0.46)] and expert websites [0.63 (0.40, 0.99)]. The findings of this study imply that the level of misconceptions is high amongst all three strata.. Expert websites and social media have a positive effect on the management of osteoarthritis. However, general practitioners and allied health workers should regularly update their knowledge using refresher courses.

## 1. Introduction

Osteoarthritis (OA) is a common aging problem. A systematic analysis of the global burden of diseases, in 2017, reported variations in the prevalence and incidence of OA among the different regions. However, an increase in the prevalence of around 10% was observed between 1990 and 2017 [1]. Age-specific prevalence for OA is reported to be 16% in individuals aged above 15 and 23% in the population aged above 40, while female-to-male ratio is 1.69:1 [1]. Amongst different joints, OA of the knee joint is the most common, with a prevalence of 16.3% in the Gulf countries [2]. The local prevalence of knee osteoarthritis in Saudi Arabia increases with age, reaching 31% in those aged 46–55 years and 61% in those aged 66–75 years [3].

The risk factors for osteoarthritis increasing with age and female gender, with the maximum burden seen between 60 and 64 years [4]. A few of the primary reasons for the increase in prevalence in this age group are menopause, obesity, and a sedentary lifestyle [1,2,5], in addition to a diet low in calcium and vitamin D [1]. Saudi Arabia has one of the highest percentages of people with obesity, physical inactivity, and an unhealthy diet [6]. Despite advancements in health education and information technology, misconceptions about osteoarthritis are common [7,8]. Misconceptions can be identified as a belief, or an idea that is based on incorrect information or that is misunderstood by people [9]. A study conducted in one of the malls in Jeddah, Saudi Arabia, found a significant difference in misconception scores due to differences in education level, sources of information, and knowing someone with osteoarthritis [7]. A common misconception among patients with osteoarthritis is that physical activity, like exercise or household work, will increase their pain and worsen their condition. Hence, the majority prefer surgical and pharmacological approaches to treat their joint pain [10]. However, evidence-based medicine proved that exercise is a crucial management approach for osteoarthritis [11]. Patients, if misguided, may end up being physically inactive, thus affecting their quality of life. Misconceptions are also associated with health-seeking behaviors, including personal interest, source of information, and type of healthcare provider. One community-based study on health behaviors related to joint pain found that physiotherapists were the most sought and trusted for receiving correct healthcare information, followed by general practitioners and expert websites [11].

Saudi culture has a close-knit family-and-friend system. Families often have an influence on the patient’s management of diseases such as OA. Previous studies from Saudi Arabia are available on the prevalence of misconceptions about OA amongst the general population [7,12]. However, they lack generalizability, as they are limited to specific populations, such as those visiting the malls or residents of a particular area. Healthcare services have expanded by including allied healthcare providers in the main workforce [13]. Similarly, with the advancement of internet sources, several people access websites and social groups to gain information about their diseases [8]. Although OA is a common disease, none of the previous studies from Saudi Arabia have tried to explore the association between health-seeking behaviors and misconceptions. For better management of OA, it is imperative to measure the association between the two. The objectives of this study were to measure the misconceptions about OA and to identify the significantly associated health-seeking behaviors.

## 2. Material and Methods

This cross-sectional study was conducted in Saudi Arabia in 2021 using an online questionnaire. The initial plan was to conduct face-to-face interviews with the patients and their attendants regarding the misconceptions and health-seeking behaviors. However, due to the COVID-19 outbreak and mobility restrictions, the data was collected using the online Google form.

Saudi Arabia is the second largest Arab country in the Eastern Mediterranean region. The country is divided into 18 small and large provinces. Rigorous methods were adopted to include participants from all the provinces. However, two provinces were not covered, due to a lack of responses. Participants were included using the non-probability sampling strategy and messages were sent out through friends, family, and acquaintances. In addition, the Google form link for accessing the survey questionnaire was circulated by utilizing all online forums, such as WhatsApp, emails, Twitter, and Telegram.

The inclusion criteria for the participants were he/she should be aged ≥18 years, can read and understand the Arabic language, and are currently residing in Saudi Arabia. In order to avoid any selection bias, healthcare providers (doctors, nurses, technicians, etc.) and Saudis living outside the country were excluded from the study. Participants were asked to give consent before proceeding to the main questionnaire. Each participant was assigned a unique identification number to maintain confidentiality. No information revealing the identity of the participants was collected. The study was approved by the Institutional Review Board (IRB) King Saud Medical City in Riyadh, Saudi Arabia, on the 2 August 2021 (21/6157/IRB), research project No. E-21-6157.

### 2.1. Data Collection Tools

An online-administered questionnaire was designed in the Arabic language and comprised three sections. The first section included sociodemographic data for ages (18–29 years, 30–49 years, and ≥50 years), gender (male and female), marital status, residence, level of education, nationality, occupation, income, and past medical history.

### 2.2. Questions on Misconceptions about OA

The second section comprised the original questionnaire developed by the study on the misconceptions about OA amongst the general population, conducted in Jeddah, Saudi Arabia [6]. The questionnaire was utilized after obtaining approval from the authors. The Jeddah study, in addition to the literature review, had adapted the questions from the validated patient knowledge questionnaire for OA, commonly known as PKQ-OA [6,13]. All the questions were translated or developed in the Arabic language. It comprised questions on the knowledge, signs and symptoms, risk factors, diagnosis, and treatment options for OA. After discussing with consultants, a question on physical activity was added, inquiring about “whether OA patients can perform simple household chores (cooking, washing, dusting)?” The response options were yes, no, or I don’t know. Each correct response was coded as 1, whereas any wrong answer or I don’t know was coded as 0. The Arabic-speaking family physician and researchers administered the face and content validity. However, we were not able to administer the criterion or concurrent validity. Pre-testing of the questionnaire was carried out to clarify any ambiguity or overlaps.

### 2.3. Health-Seeking Behaviors

The third section included questions on participants’ health-seeking behaviors for joint pain. After consulting the experts and a previously published research study [11], a set of questions was selected, adapted, and translated into the Arabic language to measure the health-seeking behaviors for joint pain. The questions included interest in acquiring information about joint pain (yes, no, only when in pain, sometimes), perceptions about acquiring treatment from different healthcare providers, and perceptions about sources of information. Different healthcare providers included general practitioners (GP), orthopedic surgeons, sports physicians, rheumatologists, podiatrists, physiotherapists, dieticians, occupational therapists, and chiropractors. The question on different sources for providing useful and trustworthy information included mobile-based health applications (for, e.g., the local health app “Sehaty”), expert websites (WebMD, drugs.com, Mayo clinic) google scholar, internet sources (for, e.g., Google search), Wikipedia, social media (WhatsApp, Twitter), news channels, printed material/magazines, family and friends. All questions had two response options, “Yes” or “No”. The Arabic-speaking family physician and researchers administered the face and content validity. However, we were not able to administer the criterion or concurrent validity. Pre-testing of the questionnaire was carried out to clarify any ambiguity or overlaps.

### 2.4. Statistical Analysis

Assuming a type-I error of 0.05, type-II error of 0.20 (power of 0.80), and a 10% difference in the misconceptions due to health-seeking attitudes, we required 600 participants. We maintained the refusal rate at >30%, after which the required sample size was 780. However, we were able to receive complete forms from 872 participants and all were included in the final analysis. Stratified analysis was conducted after dividing the participants into 3 strata; group 1 comprised 196 (22.5%) patients with physician-diagnosed osteoarthritis; group 2 comprised 249 (28.5%) participants, having joint pain (without OA); and group 3 comprised 427 (50%) participants, belonging to the general population.

The data was entered and analyzed using SPSS 26.0 version (IBM Corp., Armonk, NY, USA) statistical software. Descriptive statistics included the calculation of frequency and percentages for the categorical variables and mean and standard deviation for the continuous variables. Cross-tabulations were conducted using the Pearson chi-squared test to measure the association between sociodemographic variables, health-seeking behaviors, and misconceptions. A *p*-value of <0.05 was considered statistically significant. The effect size for the chi-squared test was calculated using Cramer’s V test [14]. The Cramer’s V effect-size (ES) interpretation was made using the following criteria: if the ES ≤ 0.2 this indicate that the results are weakly associated; if the ES > 0.2 – ≤0.6 the results are moderately associated, and if the ES > 0.6 the results are strongly associated [14].

The questionnaire section on misconceptions was converted to a categorical variable by calculating the scores for each participant by giving a score of “1” for the correct answer and a score of “0” for the wrong answer. The total score was calculated for each participant. The variable scores for the continuous misconceptions followed the normal distribution with a bell-shaped curve, with a mean value of 11. Hence, the scores were converted to a binary variable (dependent) based on the mean cutoff score of 11; participants scoring ≤11 were labeled as having inadequate knowledge (coded as 1), and those scoring >11 were labeled as having adequate knowledge (coded as 0). The independent (exposure) variables comprised various types of health care providers, ranging from general practitioners and specialists to allied health care providers such as chiropractors. The sources of information and trustworthy sources included social apps, websites, health applications, family, and friends. The responses were coded as “0” for not utilizing or trusting the source, and “1” for yes, utilizing or trusting the source. An unadjusted and adjusted logistic regression analysis was conducted to measure the odds ratio, along with a 95% confidence interval (CI) to identify the significant factors associated with misconceptions. The model was adjusted for sociodemographic factors such as age, gender, and income. Variables that were significantly associated with misconceptions or made a significant change (>10%) in the outcome variable were retained in the final model. The Hosmer–Lemeshow test was utilized to assess the model fit.

## 3. Results

The descriptive analysis identified significant differences among the three strata. The comparison of the sociodemographic characteristics of the three groups is given in Table 1. Significant differences were observed among the three groups regarding gender, age, marital status, occupation, and health status (*p* < 0.05). Around 50% of the patients in group 1 comprised elderly and middle-aged women, mostly housewives, and around 30% were diagnosed with diabetes mellitus (DM) and hypertension (HTN). Participants in group 2 mainly comprised participants between 30 and 49 years of age, working, and 20% of whom were diagnosed with HTN and 13% diagnosed with DM. The general population mainly comprised young males (18–29 years of age), 32% students, and 12% suffering from HTN and 10% suffering from DM.

No refusals were documented as the data was collected using an online Google form distributed through social media and contacts.

Table 2 shows the comparison of misconceptions about OA among the three strata. The patients in the OA group answered most of the questions correctly and scored the highest. The mean scores for the three groups were as follows: OA (12.41 ± 3.17), joint pain (10.29 ± 3.53), and the general population (10.32 ± 3.63). The mean scores were significantly different between group 1 and group 2 (*p* = 0.00) and group 1 and group 3 (*p* = 0.00); however, no significant difference was observed in the mean scores between group 2 and group 3 (*p* = 1.0). The total scores for misconceptions were converted to categorical variables, based on the average cut-off value of 11. The number of participants with inadequate knowledge (scoring < 11) in groups 1, 2 and 3 were 47%, 63.5% and 60%, respectively.

Table 3 shows the comparison of health-seeking behaviors and practices of the three strata. Around 17% of patients with OA reported that they would consult a general practitioner in the case of joint pain, whereas the percentage increased to >40% in the case of group 2 and group 3 (*p* < 0.00). Almost all the patients preferred to be seen by a consultant, with >50% of patients seeking care from an ortho surgeon (*p* = 0.02). Around 40% of patients with OA and 29% of participants with joint pain mentioned that they would consult a physiotherapist. In terms of sources of information, significant differences were observed among the three groups in the context of utilizing mobile-based health applications (*p* = 0.00), expert websites (*p* = 0.01), and Wikipedia (*p* = 0.02). In comparison with group 1, a greater number of participants from groups 2 and 3 mentioned that they would consult and trust family (*p* = 0.04) and friends (*p* = 0.07) for information on OA. Similarly, participants from groups 2 and 3 trusted mobile-based health applications (*p* = 0.01) and Wikipedia (*p* = 0.01) more, in comparison with OA patients. However, patients with OA trusted the information acquired through newspapers and tv, etc. (*p* = 0.00).

Table 4 shows the univariate and multivariate logistic regression analysis with unadjusted and adjusted odds ratio with 95% CI for the three strata. Each group was treated independently to explore the significant health-seeking behaviors associated with inadequate knowledge (misconceptions). Regression analysis for group 1 found that participants seeking care from general practitioners [3.29 (1.19, 9.16)], those utilizing mobile-based health applications [2.16 (1.03, 4.77)], and those utilizing friends [2.83 (1.08, 7.42)] as a source of information were at higher odds of having misconceptions about OA. Analysis for group 2 found that those seeking care from chiropractors [3.67 (1.02, 13.60)] and podiatrists [4.64 (1.31, 16.51)] were at higher odds of having misconceptions, whereas participants mentioning social media as a trustworthy source were at lower odds [0.16 (0.06, 0.46)] of having misconceptions. Analysis for group 3 found that working participants [2.68 (1.33, 5.41)] were at higher odds of having misconceptions, whereas participants mentioning their trust in expert websites were at lower odds of having misconceptions [0.63 (0.40, 0.99)]. The associations were significant after adjusting for age, gender and income, type of health care provider, and the sources of information.

## 4. Discussion

The results of this study imply that a high percentage of the general population and those with joint pain have several misconceptions about OA. The study also highlights the significant association between healthcare providers and friends and misconceptions about OA. In addition, the results report the positive role of social media and electronic channels in improving knowledge about OA.

Misconceptions identified by this study were reported previously by international and national studies [7,10,15]. Different types of misconceptions have been reported by the patients and the general population regarding physical activity for patients with OA [9]. Physical activity (PA) in any form is associated with better outcomes in patients with OA [16]. A systematic review conducted on barriers to PA in patients with OA found that pain and physical limitation, absence of positive PA experiences and beliefs, lack of motivation, isolation, depression, lack of social support, and lack of support by the treating physicians were the leading causes for low PA [17]) Several initiatives by the government have helped in raising awareness about healthy lifestyles amongst patients; however, for making it sustainable, it is important to involve the family and the community at each step [18]. Management of OA involves not only the patients and healthcare providers, in fact, many times, family members influence the patient’s decisions regarding the treatment [19,20]. From one perspective, this may be due to the love, care, and tendency to comfort their relatives and friends, along with being overprotective [21]. This kind of attitude can be improved by counseling the family members and explaining to them the balance between PA and taking rest. Many females are engaged in household work; however, with increasing age, they tend to rely on the younger members to do their tasks, hence becoming dependent and immobile [22]. Home-based physical activity programs are the most convenient and culturally acceptable [18,23]. A systematic review was published on the effectiveness of home-based exercise programs (HEP) for OA of the knee joint [23]. Similarly, a randomized controlled trial conducted on women with osteoarthritis from Riyadh, found that women having access to mobile-based applications were better adherents to home-based exercise programs, compared to the control group [24]. However, this study was conducted on a small number of patients from one hospital. We recommend a large-scale longitudinal study including patients from all over Saudi Arabia, which can identify the facilitators and barriers related to a home-based exercise program.

The majority of patients with OA, are initially seen by a general practitioner and later referred to a specialist. Although this is good practice, as it decreases the burden on the specialist clinic, it is important that the general physicians are well versed in the basic management guidelines and can answer patients’ queries and clarify misconceptions [25]. Studies from different parts of Saudi Arabia have shown a varying level of knowledge about OA among the general physicians, with some even showing adverse effects of physician’s advice on the patients’ health [25,26,27]. Under vision 2030, the healthcare system in KSA will be directed towards primary care [28], which makes it more important for primary care doctors and general practitioners to have correct and updated knowledge about OA. We recommend that regular refresher courses should be offered to the GPs so that misconceptions that are being passed on to their patients can be corrected, and there is also a need to improve clinician communication so that they do not inadvertently perpetuate the misconceptions that a patient already has [25].

The massive burden on the health care system has changed people’s perspective, and they find it convenient to consult alternative medical practitioners [29] and electronic channels for information and management [30,31,32,33,34]. Although the percentage of participants seeking care from alternative health practitioners such as chiropractors and podiatrist was low (10% and 13%), the association with misconceptions emphasizes the need to evaluate the knowledge and training of alternative health care practitioners. Studies are available on the role of chiropractors in treating osteoporosis; however, they should regularly update their knowledge about musculoskeletal problems, so they are able to clarify patients’ misconceptions [35,36]. Our study found that around 10–15% of the participants will consult a podiatrist for joint pain. We were not able to find the registered number of podiatrists working in Saudi Arabia. A research study from the Netherlands found that podiatrists perceived that there was an improvement in patient management after they had participated in an educational training program [37].

We found participants visiting occupational therapists (OTs) were at lower odds of having misconceptions. OTs use scientific knowledge and a holistic approach to promote a person’s ability to fulfill their daily roles and assignments [29]. Studies are available showing the positive role of OTs in primary health care and in taking care of adults; however, some studies have mentioned that OT just focuses on the patient rather than having a community approach [38]. Although the actual number of OTs registered and practicing in Saudi Arabia is not known, it is an emerging field, and with proper training and evaluation they can have a positive impact on health promotion and disease prevention for patients with OA [38].

Wrong information about a health problem can lead to adverse health outcomes. For this reason, correct and regular access to health information is important [30] Hence, the source of information must be a reliable one [39]. Even though health practitioners are more trusted, online sources such as expert websites and social media were favored, due to their easier access [31]. A study conducted on the validity of medical websites concluded that organizational/educational websites were a more reliable source of information in comparison with other sources [34]. However, a systematic review carried out in 2015 to evaluate the quality of online health information found that more than half of the articles reviewed concluded that the quality of online health information was unsatisfactory [32]. We found that acquiring information using mobile-based health applications (e.g., Sehaty) was associated with misconceptions. This may be because the application had been recently developed (during the COVID-19 outbreak) and focuses on common problems such as diabetes and hypertension. Another reason may be the limited usage of these applications by the elderly population [33].

The study findings have several implications, not only for the patient but also for the health system. The results can assist in developing specific strategies to improve the knowledge about misconceptions for the health care providers, the patients, and the general population. Patients usually take advice and listen to their family and friends; hence, involving the family/friends in the management of OA is essential. There should be regular educational sessions for caregivers. Digital health has influenced the health industry significantly. Medical-related websites and health applications should be regularized and monitored periodically so they are updated and impart correct information. The increasing aging population and the sedentary lifestyle make it imperative to inculcate physical activity into daily routines. Regular physical activity not only improves the management of OA, but also improves the quality of life.

### Strengths and Limitations

The major strength of this study is the comparison of the three strata, which has highlighted the differences in the knowledge level and the specific health-seeking behaviors. The study was the first study to measure the misconceptions of osteoarthritis at a national level in Saudi Arabia, as previously conducted studies have included a population from a specific city or a shopping mall [7,11]. The findings of our study can be generalized to other Arab countries, because the sample included participants from different cultures and ethnicities across the kingdom. This reduces any possible cultural impacts or variations due to lifestyle. By utilizing validated and well-researched questionnaires from previously published research papers, the validity and reliability of our measurement tools were assured. However, our study had some limitations. The study was conducted online during the COVID-19 outbreak; hence we had a different number of participants in each group and could not match for age and gender. In addition, it was a cross-sectional study, hence temporality cannot be established. Both information bias and selection bias cannot be ruled out.

## 5. Conclusions/Recommendations

To conclude, incorrect health-seeking behaviors for musculoskeletal problems are common, and as a consequence, misconceptions about OA are also high. General practitioners and alternate health care providers should be able to impart correct information about OA and its management. Family and friends’ advice also contributes towards having misconceptions. Protective factors included taking information from expert websites and utilizing social media for accessing health information. In the light of this study, we recommend introducing programs aimed at educating the general public about OA and encouraging them to seek out any needed information from reliable sources. This will improve their outcome and decrease healthcare costs.

## Figures and Tables

**Table 1 healthcare-11-01208-t001:** Comparison of sociodemographic factors between the three strata: patients with osteoarthritis, joint pain (without OA) and the general population, in Saudi Arabia.

Variables	Total N = 872 (%)	Group 1 Physician Diagnosed OA n = 196 (22%)	Group 2 Joint Pain (without OA) n = 249 (28%)	Group 3 General Population n = 427 (50%)	*p* Value	Cramer’s V
**Demographics**						
**Sex**						
Male	272 (31)	34 (17.3)	69 (27.7)	169 (39.6)	**<0.00**	**0.19**
Female	600 (69)	**162 (82.7)**	**180 (72.3)**	**258 (60.4)**		
**Age (in years)**						
18–29	288 (33)	8 (4.1)	81 (32.5)	**199 (46.6)**	**<0.00**	**0.31**
30–49	310 (35.6)	60 (30.6)	**108 (43.4)**	142 (33.3)		
50–70	274 (31.4)	**128 (65.3)**	60 (24.1)	86 (20.1)		
**Marital status**						
Single	302 (34.6)	10 (5.1)	84 (33.7)	208 (48.7)	**<0.00**	**0.36**
Married	570 (65.4)	186 (94.9)	165 (66.3)	219 (51.3)		
**Education**						
Diploma/University	646 (74)	150 (76.5)	181 (72.7)	315 (73.8)	0.64	0.03
Primary/Inter/Secondary	226 (26)	46 (23.5)	68 (27.3)	112 (26.2)		
**Occupation**						
Student	205 (23.4)	5 (2.6)	62 (24.5)	138 (32.4)	**<0.00**	**0.23**
Working	338 (38.9)	70 (35.9)	106 (42.6)	162 (38)		
Not working	329 (37.7)	**121 (61.5)**	82 (32.9)	126 (29.6)		
**Income**						
More than 20k	327 (37.5)	64 (32.7)	97 (39)	166 (38.9)	0.52	0.04
10k to 20k	364 (41.7)	92 (46.9)	100 (40.2)	172 (40.3)		
Less than 10k	181 (20.8)	40 (20.4)	52 (20.9)	89 (20.8)		
**Health status**						
**Diabetes Mellitus**						
(physician-diagnosed)						
Yes	143 (16.4)	73 **(37.2)**	32 (12.9)	38 (8.9)	**<0.00**	**0.31**
No	729 (83.6)	123 (62.8)	217 (87.1)	389 (91.1)		
**Hypertension**						
(physician-diagnosed)						
Yes	169 (19.4)	**67 (34.2)**	49 (19.7)	53 (12.4)	**<0.00**	**0.22**
No	703 (80.6)	129 (65.8)	200 (80.3)	374 (87.6)		

Note: Group 1 comprises patients having physician-diagnosed osteoarthritis; Group 2 comprises patients having joint pain (without OA); Group 3 comprises general population without osteoarthritis or joint pain.

**Table 2 healthcare-11-01208-t002:** Comparison of misconceptions about osteoarthritis between the three strata (patients with osteoarthritis, joint pain (without OA) and the general population) in Saudi Arabia.

Statements	Total N = 872 (%)	Group 1 Physician-Diagnosed OA n = 196 (22%)	Group 2 Joint pain (without OA) n = 249 (28%)	Group 3 General Population n = 427 (50%)	*p* Value	Cramer’s V
**Osteoarthritis is a chronic problem.**						
Correct	472 (54.1)	145 (74)	63 (25.3)	264 (61.8)	**<0.00**	0.13
Incorrect	400 (45.9)	51 (26)	186 (74.7)	163 (38.2)		
**Osteoarthritis is rare.**						
Correct	733 (84.1)	183 (93.4)	215 (86.3)	335 (78.5)	**<0.00**	**0.17**
Incorrect	139 (15.9)	13 (6.6)	34 (13.7)	92 (21.5)		
**Different joints can be affected by osteoarthritis.**						
Correct	303 (34.7)	85 (43.4)	82 (32.9)	136 (31.9)	**0.02**	**0.10**
Incorrect	569 (65.3)	111 (56.6)	167 (67.1)	291 (68.1)		
**Osteoarthritis is caused by cold, damp weather.**						
Correct	489 (56.1)	104 (53.1)	129 (51.8)	256 (60)	0.07	0.08
Incorrect	383 (43.9)	92 (46.9)	120 (48.2)	171 (40)		
**It is developed by microorganisms.**						
Correct	443 (50.8)	121 (61.7)	117 (47)	205 (48)	**0.00**	**0.12**
Incorrect	429 (49.2)	75 (38.3)	132 (53)	222 (52)		
**Pain is the only symptom of osteoarthritis.**						
Correct	469 (53.8)	88 (44.9)	134 (28.6)	247 (57.8)	**0.01**	**0.10**
Incorrect	403 (46.2)	108 (55.1)	115 (46.2)	180 (42.2)		
**Stiffness is a symptom of osteoarthritis.**						
Correct	407 (46.7)	105 (53.6)	116 (46.6)	186 (43.6)	0.07	0.08
Incorrect	465 (53.3)	91 (46.4)	133 (53.4)	241 (56.4)		
**Swelling is a sign of osteoarthritis.**						
Correct	314 (36.0)	88 (44.9)	92 (36.9)	134 (31.4)	**0.01**	0.11
Incorrect	558 (64.0)	108 (55.1)	157 (63.1)	293 (68.6)		
**Osteoarthritis can lead to loss of joint movement.**						
Correct	500 (57.3)	126 (64.3)	139 (55.8)	235 (55)	0.08	0.08
Incorrect	372 (42.7)	70 (25.7)	110 (44.2)	192 (45)		
**Genetic factors can predispose a person to OA.**						
Correct	260 (29.8)	53 (27)	81 (32.5)	126 (29.5)	0.46	0.04
Incorrect	612 (70.2)	143 (73)	168 (67.5)	301 (70.5)		
**Aging is a risk factor for osteoarthritis.**						
Correct	673 (77.2)	149 (76)	189 (75.9)	335 (78.5)	0.68	0.03
Incorrect	199 (22.8)	47 (24)	60 (24.1)	92 (21.5)		
**Men and women are equally affected by OA.**						
Correct	452 (51.8)	96 (49)	131 (52.6)	225 (52.7)	0.66	0.03
Incorrect	420 (48.2)	100 (51)	118 (47.4)	202 (47.3)		
**Physical examination and X-ray are used to diagnose OA.**						
Correct	509 (58.4)	136 (69.4)	122 (49)	251 (58.8)	**<0.00**	**0.15**
Incorrect	363 (41.6)	60 (30.6)	127 (51)	176 (41.2)		
**Blood tests are used to diagnose osteoarthritis.**						
Correct	481 (55.2)	122 (62.2)	133 (53.4)	226 (52.9)	0.08	0.08
Incorrect	391 (44.8)	74 (37.8)	116 (46.6)	201 (47.1)		
**NSAIDs can improve osteoarthritis symptoms.**						
Correct	285 (32.7)	60 (30.6)	75 (30.1)	150 (35.1)	0.32	0.05
Incorrect	587 (67.3)	136 (69.4)	174 (69.9)	277 (64.9)		
**Doing common household chores is suitable for people with osteoarthritis.**						
Correct	235 (26.9)	64 (32.7)	58 (23.3)	113 (26.5)	0.08	0.08
Incorrect	637 (73.1)	132 (67.3)	191 (76.7)	314 (73.5)		
**Some forms of exercise like swimming are suitable for people with osteoarthritis.**						
Correct	525 (60.2)	136 (69.4)	140(56.2)	249 (58.3)	**0.01**	0.10
Incorrect	347 (39.8)	60 (30.6)	109 (43.8)	178 (41.7)		
**Acid-free diets are a proven treatment for OA.**						
Correct	138 (15.8)	30 (15.3)	36 (14.5)	72 (16.9)	0.69	0.03
Incorrect	734 (84.2)	166 (84.7)	213 (85.5)	355 (83.1)		
**Physiotherapy can cause a great improvement in the symptoms of osteoarthritis.**						
Correct	634 (72.7)	145 (74)	179 (71.9)	310 (72.6)	0.88	0.02
Incorrect	238 (27.3)	51 (26)	70 (28.1)	117 (27.4)		
**Intra-articular injection by stem cell or hyaluronic acid is an effective modality for curing OA.**						
Correct	369 (42.3)	107 (54.6)	97 (39)	165 (38.6)	**<0.00**	**0.13**
Incorrect	503 (57.7)	89 (45.4)	152 (61)	262 (61.4)		
**Joint replacement surgery will be the ultimate option to relieve the symptoms of osteoarthritis**						
Correct	390 (44.7)	93 (47.4)	112 (45)	185 (43.3)	0.63	0.03
Incorrect	482 (55.3)	103 (52.6)	137 (55)	242 (56.7)		
**Categorical data for Misconceptiona about OA**						
Adequate knowledge (score ≥ 11)	365 (41.9)	103 (52.6)	91 (36.5)	171 (40)	**0.00**	**0.12**
Inadequate knowledge (<11)	507 (58.1)	93 (47.4)	158 (63.5)	256 (60)		

**Table 3 healthcare-11-01208-t003:** Comparison of health-seeking behaviors between the three strata (patients with osteoarthritis, joint pain (without OA) and the general population) in Saudi Arabia.

Variables	Total N = 872 (%)	Group 1 Physician Diagnosed OA n = 196 (22%)	Group 2 Joint Pain (without OA) n = 249 (28%)	Group 3 General Population n = 427 (50%)	*p* Value	Cramer’s V
** *If you have joint pain, from whom will you seek care?* **						
** *General practitioner* **						
No	526 (60.3)	162 (82.7)	142 (57)	222 (52)	**<0.00**	**0.26**
Yes	346 (39.7)	34 (17.3)	107 (43)	205 (48)		
** *Sports Physician* **						
No	727 (83.4)	170 (86.7)	212 (85.1)	345 (80.8)	0.12	0.07
Yes	145 (16.6)	26 (13.3)	37 (14.9)	82 (19.2)		
** *Orthopedic Surgeon* **						
No	457 (52.4)	86 (43.9)	138 (55.4)	233 (54.6)	**0.02**	0.09
Yes	415 (47.6)	110 (56.1)	111 (44.6)	194 (45.4)		
** *Rheumatologist* **						
No	648 (74.3)	139 (70.9)	189 (75.9)	320 (74.9)	0.45	0.04
Yes	224 (25.7)	57 (29.1)	60 (24.1)	107 (25.1)		
** *Physiotherapist* **						
No	561 (64.3)	119 (60.7)	177 (71.1)	265 (62.1)	**0.03**	0.09
Yes	311 (35.7)	77 (39.3)	72 (28.9)	162 (37.9)		
** *Dietician* **						
No	812 (93.1)	180 (91.8)	236 (94.8)	396 (92.7)	0.43	0.04
Yes	60 (6.9)	16 (8.2)	13 (5.2)	31 (7.3)		
** *Occupational Therapist* **						
No	814 (93.3)	185 (94.4)	234 (94)	395 (92.5)	0.61	0.03
Yes	58 (6.7)	11 (5.6)	15 (6)	32 (7.5)		
** *Podiatrist* **						
No	756 (86.7)	176 (89.8)	218 (87.6)	362 (84.8)	0.21	0.06
Yes	116 (13.3)	20 (10.2)	31 (12.4)	65 (15.2)		
** *Chiropractor* **						
No	801 (91.9)	182 (92.9)	225 (90.4)	394 (92.3)	0.58	0.04
Yes	71 (8.1)	14 (7.1)	24 (9.6)	33 (7.7)		
** *Which of the following sources will you utilize to gain more information?* **						
** *Mobile-based health applications* **						
No	535 (61.4)	139 (70.9)	154 (61.8)	242 (56.7)	**0.00**	**0.12**
Yes	337 (38.6)	57 (29.1)	95 (38.2)	185 (43.3)		
** *Expert website* **						
No	638 (73.2)	154 (78.6)	191 (76.7)	293 (68.6)	**0.01**	**0.10**
Yes	234 (26.8)	42 (21.4)	58 (23.3)	134 (31.4)		
** *Wikipedia* **						
No	731 (83.8)	177 (90.3)	203 (81.5)	351 (82.2)	**0.02**	**0.10**
Yes	141 (16.2)	19 (9.7)	46 (18.5)	76 (17.8)		
** *Google/Internet Search* **						
No	348 (39.9)	84 (42.9)	92 (36.9)	172 (40.3)	0.44	0.04
Yes	524 (60.1)	112 (57.1)	157 (63.1)	255 (59.7)		
** *Google Scholar* **						
No	776 (89.0)	181 (92.3)	222 (89.2)	373 (87.4)	0.18	0.06
Yes	96 (11.0)	15 (7.7)	27 (10.8)	54 (12.6)		
** *Social Media* **						
No	697 (79.9)	157 (80.1)	200 (80.3)	340 (79.6)	0.97	0.01
Yes	175 (20.1)	39 (19.9)	49 (19.7)	87 (20.4)		
** *Mainstream media (tv, radio, newspaper, magazine)* **						
No	820 (94.0)	181 (92.3)	236 (94.8)	403 (94.4)		
Yes	52 (6.0)	15 (7.7)	13 (5.2)	24 (5.6)	0.51	0.04
** *Friends* **						
No	651 (74.7)	159 (81.1)	186 (74.7)	306 (71.7)	**0.04**	0.09
Yes	221 (25.3)	37 (18.9)	63 (25.3)	121 (28.3)		
** *Family member* **						
No	584 (67.0)	144 (73.5)	165 (66.3)	275 (64.4)	**0.07**	0.08
Yes	288 (33.0)	52 (26.5)	84 (33.7)	152 (35.6)		
** *Which of the following sources you consider trustworthy to gain more information* **						
** *Mobile-based health application* **						
No	385 (44.2)	104 (53.1)	111 (44.6)	170 (39.8)	**0.01**	**0.11**
Yes	487 (55.8)	92 (46.9)	138 (55.4)	257 (60.2)		
** *Expert website* **						
No	623 (71.4)	138 (70.4)	187 (75.1)	298 (69.8)	0.31	0.05
Yes	249 (28.6)	58 (29.6)	62 (24.9)	129 (30.2)		
** *Wikipedia* **						
No	779 (89.3)	187 (95.4)	217 (87.1)	375 (87.8)	**0.01**	**0.11**
Yes	93 (10.7)	9 (4.6)	32 (12.9)	52 (12.2)		
** *Google/Internet Search* **						
No	598 (68.6)	126 (64.3)	172 (69.1)	300 (70.3)	0.32	0.05
Yes	274 (31.4)	70 (35.7)	77 (30.9)	127 (29.7)		
** *Google Scholar* **						
No	768 (88.1)	178 (90.8)	216 (86.7)	374 (87.6)	0.38	0.05
Yes	104 (11.9)	18 (9.2)	33 (13.3)	53 (12.4)		
** *Social Media* **						
No	796 (91.3)	173 (88.3)	226 (90.8)	397 (93)	0.15	0.07
Yes	76 (8.7)	23 (11.7)	23 (9.2)	30 (7)		
** *Mainstream media (tv, radio, newspaper, magazine)* **						
No	829 (95.1)	178 (90.8)	243 (97.6)	408 (95.6)	**0.00**	**0.11**
Yes	43 (4.9)	18 (9.2)	6 (2.4)	19 (4.4)		
** *Friend* **						
No	720 (82.6)	164 (83.7)	214 (85.9)	342 (80.1)	0.14	0.07
Yes	152 (17.4)	32 (16.3)	35 (14.1)	85 (19.9)		
** *Family member* **						
No	621 (71.2)	144 (73.5)	172 (69.1)	305 (71.4)	0.59	0.03
Yes	251 (28.8)	52 (26.5)	77 (30.9)	122 (28.6)		

**Table 4 healthcare-11-01208-t004:** Univariate and multivariate analysis showing association between health-seeking behaviors and misconceptions in patients with osteoarthritis, joint pain (without OA) and general population in Saudi Arabia.

Variables	Unadjusted Odds Ratio (95% CI)	Adjusted Odds Ratio (95% CI)
**Group 1 Patients with OA**		
** *General practitioner* **		
No	**1.0**	**1.0**
Yes	**3.23 (1.45, 7.20)**	**3.29 (1.19, 9.16)**
** *Mobile-based health application* **		
No	1.0	**1.0**
Yes	1.64 (0.88, 3.05)	**2.16 (1.03, 4.77)**
** *Friends* **		
No	1.0	**1.0**
Yes	**2.09 (1.0, 4.35)**	**2.83 (1.08, 7.42)**
**Group 2 People with joint pain (without OA)**		
** *Chiropractor* **		
No	1.0	**1.0**
Yes	2.35 (0.85, 6.53)	**3.67 (1.02, 13.60)**
** *Podiatrist* **		
No	1.0	**1.0**
Yes	2.15 (0.89, 5.21)	**4.64 (1.31, 16.51)**
** *Occupational therapist* **	1.0	
No		**1.0**
Yes	0.64 (0.22, 1.83)	**0.24 (0.06, 0.99)**
** *Social media* **		
No	**1.0**	**1.0**
Yes	**0.27 (0.11, 0.67)**	**0.16 (0.06, 0.46)**
**Group 3 General population**		
** *Occupation (n = 870)* **		
Student	**1.0**	**1.0**
Working	**2.31 (1.44, 3.70)**	**2.68 (1.33, 5.41)**
Housewives/Retired	1.57 (0.96, 2.55)	2.04 (0.98, 4.23)
** *Expert website* **		
No	**1.0**	**1.0**
Yes	**0.63 (0.42, 0.95)**	**0.63 (0.40, 0.99)**

## Data Availability

Data can be shared on specific request sent to the corresponding author.

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
