# Peer review of "Health-Seeking Behaviors and Misconceptions about Osteoarthritis in Patients and the General Population in Saudi Arabia"

_healthcare, 2023, doi:10.3390/healthcare11091208_

Round 1

Reviewer 1 Report

1. In abstract section, I suggest that the rationality of this investigation and the description of the conclusion can be appropriately added to avoid the excessive content of the result section.

2. Please add the rationality and source description of the questionnaire.

3. The content of research methods is too single, so the description of research variables can be added.

4. A description of the significance of the study can be added to the results section. And please discuss whether the findings are generalizable.

5. Language quality needs to be further improved.

6. In methods section, you can cite the article PMID: 34774463, PMID: 33946978 to enhance the explaination of the statistical analysis. You can also refer to PMID: 36388298 to explain the method of Cross section study.

Reviewer 2 Report

I thank the editors for choosing me to review this manuscript. I congratulate the authors for the realization of this interesting article. I think some fixes are needed which I will list below: MATERIALS AND METHODS - Assuming a type-I error of 0.05, type-II error of 0.20 (power of 0.80), and 10% difference in the misconceptions due to health-seeking attitudes, we required 600 participants. Keeping the refusal rate at >30%, interviews were conducted with 872 participants....I would put this whole paragraph in the statistical analysis section. - were the only inclusion criteria knowledge of the Arabic language and age >18? are few, you should provide other inclusion criteria. Same thing with regard to the exclusion criteria. - why were health professionals excluded from the study? motivate RESULTS - The sample comprised of 872 participants, further divided into 3 groups based on their musculoskeletal health. Group 1 comprised of 196 (22.5%) patients with physician diagnosed osteoarthritis; group 2 comprised of 249 (28.5%) participants, having joint pain only (without OA); group 3 comprised of 427 (50%) participants from the general population.... you did not specify this sampling method used in the methods section. Add. - Misconceptions identified by this study were reported previously by International and national studies [11, 12, 15]. Similar to previous study [3], OA was more prevalent amongst females compared to the males (27% vs 12.5%). It is reported that synovial fluid exosomal microRNAs content is altered in patients with OA and these changes are gender specific [17]. Decrease in estrogen levels in females after menopause is found to be associated with changes in the synovial fluid in the joints [17]. We observed that with aging, the percentage of women reporting OA increased significantly (increased from 5% to 28%). Previous studies have found that average age for menopause in Saudi women is around 49-50 years [18]. Hence the hormonal and bone changes occurring around menopause can explain the increase in OA prevalence..... I don't think this whole paragraph has anything to do with the scope of your study. - in the strengths of your study you should indicate what makes your study unique and different from other previous studies that have analyzed the topic and found the same results

Reviewer 3 Report

1. Need more explanation as why score>11 was selected as the cutoff for the outcome.

2. In the methods section, the authors mentioned the sample size of 872 was selected based on >30% refusal rate. However, in the results section, no refusal was mentioned.

3. For descriptive results and logistic regression, what's the rational of reporting results separately by three groups? Why not consider the grouping as one covariate?  Then would it be more meaningful to compare results across groups?

Round 2

Reviewer 3 Report

The authors have addressed the previous comments.

Author Response

RESPONSE TO EDITORS COMMENTS

Manuscript ID number:  healthcare-2238933

Health seeking behaviors and misconceptions about osteoarthritis in patients and the general population in Saudi Arabia

We appreciate the Editor’s comments and believe the revisions we have made in response in the attached version of the manuscript have improved the quality of the manuscript.  Below we indicate the changes made and where in the manuscript they were made in response to these comments.

Comments & Reply

-Indicate whether the translation of the PKQ-OA questionnaire and the "Health Seeking Behaviors" questionnaire from English to Arabic was validated

Reply: The translation of the PKQ-OA and the “Health Seeking Behaviors” underwent through face and content validity. However, the concurrent validity was not conducted. Please refer to page no 3, line 142-145 and line 162-167.

- calculate and report on effect size and confidence interval for all statistical tests performed.

Reply:  Suggestion taken, we have calculated the effect size for the chi -square test and have mentioned the values accordingly. Please refer to Table 1, 2 and 3.

In the Table no 4, the unadjusted and adjusted odds ratio along with confidence interval were calculated for each of the groups independently.
